# Looking beyond Self-Protection: The Eyes Instruct Systemic Immune Tolerance Early in Life

**DOI:** 10.3390/brainsci13091261

**Published:** 2023-08-30

**Authors:** Horacio Villafán, Gabriel Gutiérrez-Ospina

**Affiliations:** 1Programa de Doctorado en Ciencias Biológicas, Unidad de Posgrado, Circuito de Posgrados, Ciudad Universitaria, Edificio D, 1^er^piso, Coyoacán, Ciudad de México 04510, Mexico; 2Departamento de Biología Celular y Fisiología, Instituto de Investigaciones Biomédicas, Universidad Nacional Autónoma de México, Ciudad de México 04510, Mexico; 3Coordinación de Psicobiología y Neurociencias, Facultad de Psicología, Universidad Nacional Autónoma de México, Ciudad de México 04510, Mexico; 4Department of Zoology and Physiology and Wyoming Sensory Biology Center of Biomedical Research Excellence, University of Wyoming, Laramie, WY 82071, USA

**Keywords:** eye enucleation, skin contact hypersensitivity, blindness, neuro–immune interactions, skin inflammation, T regulatory lymphocytes, skin–brain axis

## Abstract

The eyes provide themselves with immune tolerance. Frequent skin inflammatory diseases in young blind people suggest, nonetheless, that the eyes instruct a systemic immune tolerance that benefits the whole body. We tested this premise by using delayed skin contact hypersensitivity (DSCH) as a tool to compare the inflammatory response developed by sighted (S) and birth-enucleated (BE) mice against oxazolone or dinitrofluorobenzene at the ages of 10, 30 and 60 days of life. Adult mice enucleated (AE) at 60 days of age were also assessed when they reached 120 days of life. BE mice displayed exacerbated DSCH at 60 but not at 10 or 30 days of age. AE mice, in contrast, show no exacerbated DSCH. Skin inflammation in 60-day-old BE mice was hapten exclusive and supported by distinct CD8^+^ lymphocytes. The number of intraepidermal T lymphocytes and migrating Langerhans cells was, however, similar between S and BE mice by the age of 60 days. Our observations support the idea that the eyes instruct systemic immune tolerance that benefits organs outside the eyes from an early age. The higher prevalence of inflammatory skin disorders reported in young people might then reflect reduced immune tolerance associated with the impaired functional morphology of the eyes.

## 1. Introduction

The eyes are the premier visual sensory organs. They, however, provide themselves with immune tolerance when challenged with foreign antigens [1,2,3,4,5,6,7,8]. This property emerges from a complex network of interactions established among the cellular and acellular components within the eyes and between the eyes and the lymphoid organs across the body [9,10,11,12,13,14,15,16,17,18,19].

Many assure that the immune tolerance induced by the eyes is intended to avoid intraocular damage [20]. However, the fact that young people with congenital or acquired after birth (fever, post-infection, accident, or surgery) blindness frequently develop inflammatory skin diseases [21,22], suggests that the eyes may help secure immune tolerance across the body, that is, beyond eye boundaries. In support of this notion, it is known that altering the photoperiod deteriorates delayed skin contact hypersensitivity (DSCH) in rodents [1,2,8], light entering the eyes regulates immune function through an assortment of mechanisms in a variety of animals, including humans [23], corneal nerve ablation disables immune tolerance induced by mucosal surfaces after promoting the generation of contra suppressor cells [10,11,12,13,14,15,16,17], and neonatal enucleation leads to increased macrophage phagocytic activity in the liver of adult rats [24]. It is also common to observe oculocutaneous manifestations preceding multisystem inflammatory disorders [25,26].

In mice, adult immune responses are conditioned early in postnatal development [27,28,29]. Immune tolerance, on the other hand, is set up by regulatory T lymphocytes early in life [30,31]. It is known that immune tolerance induced by the eyes promotes the formation of regulatory T lymphocytes [4,5,6,7,8,9,10,11,12,13,14,15,16,17,18,19,20,21,22,23,24,25,26,27,28,29,30,31,32]. Thus, it could be that the systemic immune tolerance instructed by the eyes and the benefits it brings to the body are primed early in life. In this work, we tested these premises by evaluating DSCH in adult mice that were enucleated at birth or at 60 days of age. Mice enucleated at birth but not enucleated at adulthood showed enhanced DSCH. These observations suggest that the eyes prime systemic immune tolerance at an early age. Once this happens, such tolerance becomes permanent, providing lasting benefits to the body, even if the eyes are absent later in life. Thus, upcoming clinical studies must assess whether the systemic immunological tolerance of young blind people with a higher-than-average prevalence of inflammatory skin disorders is indeed impaired and whether adult individuals with late-onset blindness have immune tolerance relatively preserved.

## 2. Materials and Methods

### 2.1. Animals

Eight- to ten-week-old, outbred, timed, pregnant CD1 female mice were housed in barriers in individual cages and maintained at temperature (21–23 °C), humidity (50%), and light (day/night regular cycles, 7:00 light on/19:00 light off) controlled rooms located in the Unidad de Modelos Biológicos, Instituto de Investigaciones Biomédicas (IIB), Universidad Nacional Autónoma de México (UNAM). Pregnant mice had free access to sterilized mouse chow and clean water. The cages had sterilized pelleted hardwood bedding that was changed twice weekly. Upon delivery, litters were divided into two groups. The first cohort provided mice used to perform the developmental study. Mice from the second cohort were raised until adulthood. In each case, the number of pups was adjusted to eight per litter, keeping the original sex ratio constant for each one. Pups were weaned at the age of 21 days. All males were housed in groups of five after weaning and kept in the animal facility as already described. The experiments were conducted only on male pups. Half of the pups from the litters dedicated to the development study underwent bilateral enucleation six hours after birth. The other half remained sighted (S) but underwent the same manipulations as their BE siblings, except for surgery. On the other hand, pups dedicated to adult experiments were kept sighted until 60 days of age. Here too, half of the adult males (A) underwent bilateral enucleation on postnatal day 61; the other half were kept in sight.

For developmental studies, S and BE pups were cryo-anesthetized after being placed on crushed ice for up to 12 min; this time, it interrupts neural transmission and has no impact on the further development of the pups. Regularly, analgesia for hypothermia induced by this method lasts approximately 10 min, enough time to conduct the enucleation bilaterally. At the end of the procedure, each pup was placed on a thermal blanket until it fully regained movement, color, and temperature. Full recovery usually occurred after about 30 min, after which time the pups were returned to their mothers. Pups were always monitored to avoid overheating and/or rapid overheating, as the latter could cause tissue damage. We observed the behavior of each mother after returning her litters to her; no evidence of rejection was observed in any case. Finally, bilateral enucleation in adult mice was performed under anesthesia induced with ketamine and xylazine; sighted peers of the same age underwent a similar manipulation but the surgery. Animal handling and experimental procedures conformed to the rules and recommendations provided by the NIH Guide for the Care and Use of Laboratory Animals (8th Edition) and were reviewed and approved by the Comité Para el Cuidado y Uso de Animales de Laboratorio IIB, UNAM (Protocol #80).

### 2.2. Enucleation

Each pup was scrubbed once with a sterile gauze pad moistened in a 70% isopropyl alcohol USP solution. The eyelids and orbital region of each pup’s face were cleansed three times with a povidone-iodine solution with the aid of cotton swabs. Pups were transferred to and placed in the surgical field. An incision along the inter-palpebral fissure was made using a sterile stainless-steel scalpel blade (#11). The eyelids were carefully separated and the eye exposed. Each eye was freed from the musculature and grabbed from behind by carefully sliding a fine-tip precision tweezer from the edge to the back of the orbit. After detaching the musculature and the optic nerve, the eye was carefully pulled outward, plucking it from the orbit while avoiding squeezing it during the procedure. Commonly, bleeding is minute and easily controlled with light compression. The blood clot effectively sealed the incision. The procedure was repeated on the other side of the face. Sterile operating room sheets were used to cover the surgical field, but none were used to cover up the pups. The procedure was performed under stereomicroscopic guidance and followed aseptic protocols during the execution of the operation. There was no postsurgical need to use antibiotics or analgesics; morbidity and mortality are extremely low, especially if the procedure is performed by experienced staff. Adult bilateral enucleation was performed following the guidelines already described. No palpebral surgical incision was made. Eye extraction was conducted as noted. Bleeding was controlled by light compression, and ophthalmic ointments having lidocaine (lidocaine hydrochloride jelly USP 2%; Akorn, Lake Forest, IL, USA) and erythromycin (0.5% Bausch and Lomb, Rochester, NY, USA) were used to avoid infections and reduce discomfort after surgery. Lastly, hypothermia was prevented by warming the mice with a thermal blanket until recovery. Adult S and AE mice were returned to their home cages upon recovery.

### 2.3. Ear Swelling Test

Even though we do not have reliable data on the natural frequency of inflammatory skin disorders in mice, it is relatively common to induce DSCH in laboratory mice [33,34,35,36,37]. In mice, DSCH is elicited experimentally after challenging ear skin with low-molecular-weight haptens, which have been used to sensitize abdominal skin days before [38]. Here, we sensitized the abdominal skin with 100 μL of 4-ethoxymethylene-2-phenyl-2-oxazolin-5-one (oxazolone; OXA; 2% in 100% ethanol; SIGMA-ALDRICH, St. Louis, MO, USA) or with 25 μL of 2,4-dinitrofluorobenzene (DNFB; 0.5% diluted in acetone/olive oil; 4:1). After five days of priming, mice were challenged by stroking OXA (1%) or DNFB (0.2%) on the left ear. The right ear of each mouse, used as an internal control, was stroked with ethanol or acetone-olive oil, correspondingly. Although both haptens induce DSCH, they act through different immune pathways and cytokine cascades [39]. To assess the specificity of the immune response for each hapten, 60-day-old S and BE mice were sensitized with OXA or croton oil (10 μL; 0.8% diluted in acetone; SIGMA-ALDRICH) and challenged five days later with DNFB or croton oil, respectively. For mice “sensitized” and “challenged” with croton oil, the right ear was defied with acetone; croton oil elicits an irritant response only [40]. OXA-DSCH tests were performed in 60-day-old S and BE mice and 120-day-old S and AE mice. DNFB-DSCH tests were performed in 10-, 30-, and 60-day-old mice. In every case, the thickness of both ears was measured with a spring-loaded micrometer every 24 h for five days, and the data are reported and plotted as the difference of the thickness between the left and the right ears.

### 2.4. Ear Cytokine Inflammatory Profile

DSCH features local increments of pro-inflammatory cytokines. We assessed whether the profile of proinflammatory cytokines associated with exacerbated DSCH in 60-day-old BE mice was higher relative to S mice after priming and OXA challenge (*n* = 3/group in a single assay). The mice were sacrificed 24 h after performing the test using CO_2_. The ears were separated from their heads and mechanically homogenized in deionized water at pH 7.4 (titrated, if needed, with NaOH 0.1 N) at 4 °C. The homogenates were then frozen on dry ice and thawed at 37 °C. Subsequently, they were sonicated and centrifuged at 14,000 rpm for 5 min at 4 °C. The supernatants were collected, their protein content quantified (Bradford’s assay; Bio-Rad, Hercules, CA, USA), and the concentration of pro-inflammatory cytokines was estimated spectrophotometrically (at λ 450 nm) by using the Multi-Analyte ELISArray Kit, following the manufacturer’s instructions (SABiosciences, Frederick, MD, USA).

### 2.5. Ear Skin Histology

DSCH is a process mediated by memory T lymphocytes. Here, lymphocyte infiltration was initially corroborated in Hematoxylin–Eosin (HE) stained material, followed by immunostaining against the co-receptor CD8, a membrane glycoprotein featured by memory T lymphocytes [41]. For HE (*n* = 6/group, three replicates comprising two mice each), OXA-sensitized and challenged 60-day-old S and BE mice were euthanized at the peak of the inflammatory response with pentobarbital (40 mg/kg body weight). Each mouse was perfused through the heart with saline solution (0.15 M) followed by buffered paraformaldehyde (4%; phosphate buffer 0.1 M, pH 7.4; PB). The left and right ears were detached from their heads and immersed for 2 h in the same fixative. Small blocks of tissue (3 mm^3^) were crafted from control and challenged ears. These blocks were dehydrated in alcohol, defatted in xylene, and embedded in paraffin. Microtome sections (5 μm thick; 10 sections per mouse) were mounted on glass slides (Super Frost, Thermo Fisher Scientific, Waltham, MA, USA), dried, dewaxed, rehydrated and stained with HE, following a conventional protocol sequence. Slides were finally dehydrated in alcohol, cleared in xylene, and mounted with Cytoseal 60 (Thermo-Shandon (Cambridge, UK), Richard Allan Scientific (Canton, MI, USA), and CTR Scientific (Monterrey, Mexico). For immunohistochemistry (*n* = 9/group, three replicates comprising three mice each), mice were euthanized, and their ears were dissected and processed as above. Ear tissue blocks were rapidly frozen in 2-methyl butane pre-chilled with dry ice. Cryostat sections (10 μm thick; 10 sections per mouse) were mounted on gelatin-coated glass slides, air-dried and stored at −70 °C until use. On the day of the experiment, the sections were fixed in acetone at −20 °C for 15 min, washed in PB and pre-incubated with blocking serum (3% goat serum diluted in PB) for three hours at room temperature. The sections were incubated with primary antibodies raised in rats against mouse CD8 (1:100; Chemicon, Temecula, CA, USA) diluted in blocking solution (goat serum diluted 1:1000 in PB) for 18 h at 4 °C. After three washes in PB at room temperature, sections were incubated with secondary antibodies raised in donkey against rat IgG (1:1000 diluted in PB; Invitrogen, Carlsbad, CA, USA), labeled with Alexa Fluor 594, for 2 h at room temperature. After three washes in PB, slides were cover-slipped with an anti-fade mounting medium (Vector Laboratories, Newark, CA, USA). All this material was used to describe qualitatively the histological features of the skin and corroborate (or not) the presence of immune infiltrates in OXA-challenged (left) or vehicle-exposed (right) ears. The images were captured (60×) and digitized by using an upright, bright-field Nikon Optiphot 2 microscope equipped with a Nikon Coolpix 900 digital camera (Nikon Instruments Inc., Melville, NY, USA.

### 2.6. Substance P Antagonization

DSCH involves the exchange of immune cells back and forth between skin-challenged sites and the nearby lymph nodes. The mobilization of specific memory T cells from the lymph nodes to the skin during DSCH is favored by the substance P released from sensory neurons [40,42,43,44,45,46,47,48]. To evaluate whether the exacerbation of DSCH in BE mice indeed involved immune memory, we assessed whether ear swelling could be prevented or attenuated in S and BE mice after antagonizing the actions of substance P in the auricular lymph node. Accordingly, sixty-day-old S and BE mice were OXA-sensitized and challenged four weeks later (in a single assay with 3 animals per group). The left auricular ganglion was bathed, by subcutaneous injection, with a solution containing cis-2-(Diphenylmethyl)–*N*–[(2-iodophenyl) methyl]-1-azabicyclo [2.2.2] octan-3-aminutese oxalate salt (L-703,606; 5 mg/mL in 120 μL saline; SIGMA-ALDRICH). The drug was administered thirty minutes before applying the challenge. The right auricular lymph node was bathed with saline only. Ear thickness was estimated 24 h later.

### 2.7. Dendritic Epidermal T Cells (DETCs) Number

Local DETCs downregulate skin inflammatory responses [49,50,51,52,53,54]. A differing number of them could explain the exacerbation of DSCH in 60-day-old BE mice. Therefore, 60-day-old S and BE mice (*n* = 9/group, three experiments with three animals each) were sacrificed to explore whether the number of DETCs decreased in the ear skin after enucleation at birth. After sacrifice, the ears were clipped from each mouse head and separated into medial and lateral flaps. The outer flap was incubated in a solution containing 0.5 M EDTA for 2–3 h at room temperature until the epidermis separated from the dermis. The epidermal sheets were washed in PB, fixed in 100% acetone at −20 °C for 15 min, and washed again in PB. They were incubated first in blocking serum (3% bovine serum albumin diluted in PB; BSA) for 2 h at room temperature, and then with primary antibodies raised in hamsters against mouse CD3ε (145-2C11; Chemicon), diluted 1:1000 in blocking serum, for 18 h at 4 °C. After 3 washes in PB, the epidermal sheets were incubated with biotin-labeled goat anti-hamster IgG secondary antibodies (1:1000; Vector) for 2 h at room temperature. Once again, samples were washed three times in PB and incubated with biotinylated peroxidase diluted in PB for three hours at room temperature (Vectastain Elite ABC Kit, Vector Laboratories). Lastly, after three additional washes, epidermal sheets were incubated with 3,3-diaminutesobenzidine and H_2_O_2_. The reaction was stopped one minute later by thoroughly washing with PB. Epidermal sheets were mounted onto gelatin-subbed slides, dried at room temperature, and cover-slipped with Cystoseal 60. The stained samples were used to capture digital images across epidermal sheets. Ten fields were taken randomly from each left and right ear per animal at 60×. The density of intraepithelial CD3ε positive T lymphocytes was estimated by counting manually the number of cells in 9 mm^2^. Images were captured and digitized by using an upright Nikon Optiphot 2 bright-field microscope equipped with a Nikon Coolpix 900 digital camera.

### 2.8. Skint1 mRNA Expression

*Skint1* is a protein that positively selects intraepithelial T lymphocytes while in the thymus. It also helps target these cells to the epidermis during fetal development [55]. *Skint1* is expressed by keratinocytes [56,57]. Because, on the one hand, keratinocytes in BE mice are hypertrophic, and on the other hand, DETCs colonize the skin shortly after birth [56], we thought that Skint1 expression might differ between S and BE mice. Sixty-day-old mice of both groups (*n* = 6/group, two experiments with two or three animals each) were euthanized by using CO_2_. Their ears were separated from their heads and preserved in TRIzol (Thermo Fisher Scientific). The samples were incubated with DNase, RNase-free, as advised by the manufacturer (Invitrogen). RNA was reverse transcribed with SuperScript II (42 °C for one hour). cDNA was amplified with Taq DNA polymerase (Invitrogen) using the forward primer: cttcttcagatggtcacagca and the reverse primer: cctgttagagggttctga (*Skint1* GenBank entry: EF494889.1). Polymerase chain reaction was performed during 30 cycles (95 °C/30 s, 55 °C/60 s, and 72 °C/60 s) using a T100 Thermal Cycler (Bio-Rad Laboratories, Ciudad de México, México).

### 2.9. Analyses of Langerhans’ Cells Migration by Flow Cytometry

Langerhans cells play a central role in modulating the adaptive lymphocytic immune response during the DSCH sensitization phase [58,59,60,61,62]. Facilitated migration of Langerhans cells from the skin to the lymph nodes could explain the exacerbation of DSCH in BE mice, especially since OXA-induced DSCH reduces lymphatic drainage while enhancing the adaptive immune response [63]. Then, the thorax and abdomen of 60-day-old S (*n* = 5 per time point) and BE (*n* = 8 per time point) mice were shaved and varnished with 200 μL of fluorescein isothiocyanate isomer 1 (FITC; 1%; SIGMA-ALDRICH) dissolved 1:1 in acetone: dibutylphtalate [64]. Mice were euthanized with CO_2_ 24, 48, 72 and 96 h after FITC application. The inguinal, brachial, and axillary lymph nodes of each mouse were obtained, pulled for minutes, and incubated in a solution containing collagenase (1 mg/mL) and DNAse (0.02 mg/mL) in RPMI 1640 (Roswell Park Memorial Institute; Gibco Thermo Fisher Scientific, Ciudad de México, México) for 20 min at room temperature; EDTA (0.1 M) was added 5 min before the end of the incubation to inactivate collagenase. Disaggregated lymph nodes were filtered through 70-μm-pore-diameter sieves. The FITC-labeled cell suspension was recovered and washed in phosphate-buffered saline (15 mM pH 7.4) supplemented with fetal bovine serum (2%) and NaN_3_ (0.02%) (PBS-BSA), pelleted by centrifugation (1800 rpm) at 4 °C for 10 min, and incubated with normal mouse serum (obtained from the animal facility at IIB) diluted 1:20 in PBS-BA for 30 min at 4 °C. FITC-labeled lymphoid cells were identified using PE hamster anti-mouse CD11c (1:100; BD Biosciences, Franklin Lakes, NJ, USA) and PE/Cy5 rat anti-mouse I-A/I-E (MHC-II; 1:100; BioLegend, San Diego, CA, USA) monoclonal antibodies for 20 min at room temperature. After washing with PBS-BSA, cells were fixed with 1% buffered paraformaldehyde for 20 min and washed. Cells were permeabilized with saponin (0.5%) diluted in PBS-BSA for 15 min at room temperature and incubated with Alexa Fluor 647 Rat anti-mouse CD207 antibody (1:100; eBioscience, San Diego, CA, USA) diluted in PBS-BSA. They were analyzed on a FACSCalibur cytometer (BD Biosciences, Ciudad de México, México). Data were analyzed with Cell Quest and FlowJO software both by BD Bioscience.

### 2.10. Statistics

Experiments and data collection were conducted double-blind after having the mice randomly coded by people (non-participants) different from those involved in conducting either the experiments or the data collection and analyses. The validity of the null hypothesis for the mean values obtained from the parameters considered was evaluated by using pairwise, two-tailed Student’s *t*-tests (contact hypersensitivity analysis, DETCs counting, and FITC+ Langerhans’ cells migration) or ANOVA (pro-inflammatory cytokine measurements) by using GraphPad Prism 6.05 and 8.1.2 Software and Origin 9.1 Software. A normality test was performed on the data set that corresponds to DETC counts. Values of *p* < 0.05 were considered statistically significant. Results are expressed as mean ± SEM. Graphs were made with both software programs mentioned before.

## 3. Results

### 3.1. Skin Contact Hypersensitivity Is Exacerbated in Adult Mice Enucleated at Birth

We postulated that systemic immune tolerance provided by the eyes and the benefits it brings to the body are primed early in life. This concept was verified by combining birth or adult bilateral enucleation with DSCH assessment at different ages. Accordingly, 60-day-old BE mice developed an exacerbated DSCH response after being challenged with either OXA (Figure 1a and Figure 2a) or DNFB (Figure 2). For DNFB, this response was unseen at PD10 or PD30 (Figure 2) and was absent in AE mice when sensitized and challenged with OXA 60 days after enucleation (Figure 3). Thus, these results circumstantially support the prediction made and further suggest that once DSCH is strengthened and consolidated by the age of 60 days, it endures.

Ear swelling during DSCH must be accompanied by inflammation. We then corroborated that inflammation does occur in the ears of OXA-sensitized S and BE mice. First, we showed that the ears in both groups of mice increased the availability of proinflammatory cytokines at the peak of DSCH when challenged (Figure 1b). Although cytokine availability varied individually within and between groups of mice, as a population, BE mice tended to exhibit higher concentration values for nine of the twelve proinflammatory cytokines tested (Figure 1b). Definitive results, nonetheless, will require increasing the sample size. Despite this shortcoming, the inflammatory cytokine profile observed in BE mice is comparable to the one previously reported for delayed-type hypersensitivity [65,66,67]. This circumstance supports the idea that the exacerbated DSCH in 60-day-old BE mice involves adaptive immunity and does not simply result from a non-inflammatory edema. 

Ears from OXA-sensitized S (Figure 1d) and BE (Figure 1f) mice not only showed proinflammatory cytokine profiles but also displayed lymphocyte infiltrations after being challenged with OXA (Figure 1c) but not with ethanol (Figure 1e). Qualitatively, the infiltrates appear “denser” in BE mice and have abundant lymphocyte clumps (compare dermal infiltrates in Figure 1d,f), a histological landscape suggesting that memory T cells were mobilized and recruited in the challenged ear [66,67]. In support of this possibility, we found that CD8^+^ T lymphocytes are part of the infiltrates (Figure 1g) and that the substance P antagonist L-703,606 prevented DSCH in OXA-challenged, sensitized, 60-day-old BE mice (Figure 4). Although we did not document the arrival of CD4^+^ helper/regulatory T cells to OXA-sensitized ears, it is known that IL12, IL6 and IL4 cytokines, which were found to be elevated in OXA-sensitized BE mice, govern CD4^+^ lymphocyte dynamics and differentiation during skin contact hypersensitivity reactions [68]. So, it is reasonable to expect that they were also present in the lymphocyte infiltrations of OXA-sensitized/challenged ears.

Lastly, DSCH induced by OXA or DNFB in 60-day-old S and BE mice was specific since no DSCH was observed when OXA and DNFB-sensitized S or BE mice were cross-challenged with these haptens (Figure 5a) or when they were sensitized and challenged with croton oil (Figure 5b). In addition, the fact that the kinetics of the response for each hapten in both mouse groups differed (compare Figure 1a and Figure 4c) supports the specificity of DSCH responses observed and agrees with the conclusions that OXA and DNFB induce DSCH through distinct mechanisms [39,67,69,70].

### 3.2. Shifts in DETC Density Do Not Explain Exacerbated DSCH Response in BE Mice

Atopic dermatitis features epidermal thickening [71,72,73,74] prompted in part by the interplay of neurotrophins [75,76,77,78] and numerous cytokines [79,80]. Here, we found the epidermis thickened (Figure 1c–f and Figure 6a,b; see also [81] for a similar result in rats) and some of the proinflammatory cytokines involved in regulating keratinocyte growth and differentiation (e.g., IL4, IL6 and IL17) upregulated two months after enucleation at birth. In a previous study, neurotrophin-3 was also found to be increased in the whisker muzzle of BE rats [82]. These results suggest that, in the absence of eyes from early life, the rodent skin develops a condition characterized by a deregulated immune response that can be evidenced by sensitizing and challenging birth-enucleated rodents with OXA and/or DFBN. The eyes then provide information from an early age that helps the skin regulate its immune responses in adulthood [83].

DETCs help maintain skin homeostasis, modulate skin immune responses [38,49,50,51,52,53,54,72,80,83], and balance keratinocyte differentiation and proliferation [57,84,85]. Since DETCs colonize the skin early in life [56], we thought that enucleation at birth could alter DETC numbers in the skin, leading to an exacerbated DSCH response and a thickened epidermis later in life. Estimates of the CD3ε/Vγ5^+^ DETC basal density in epidermal sheets of 60-day-old S and BE mice revealed, nonetheless, the existence of two similar phenotypes in both experimental groups of mice. Roughly half of S and BE mice display DETC numbers below average values. The rest of them had DETC numbers above average values (Figure 6c–e). These phenotypes segregated more clearly after enucleation at birth (Figure 6d1–e). Phenotype diversity within and across experimental groups of mice was also observed when Skint1, a gene whose protein product promotes DETC positive selection in the thymus and skin homing [55,56,57,86,87], was amplified in ear skin samples obtained from 60-day-old S and BE mice. Skint1 mRNA was convincingly amplified in two of six S mice (Figure 6f exp1, exp2). A weak amplification was obtained in one of six S mice (Figure 6f exp1) and in two of six BE mice (Figure 6f exp1, exp2). We were unable to amplify Skint1 in the ear samples of the remaining mice in both groups (Figure 6f exp1, exp2). At the present time, the results described do not allow us to infer the role of DETCs in DSCH, whether it is exacerbated or not, because all 60-day-old S and BE mice sensitized and challenged with OXA or DNFB developed a robust DSCH response with relatively small intragroup variations. Nevertheless, we think DETCs are likely involved as they produce cytokine 17a, a protein that tends to be elevated in BE mice. Future studies must be designed to dissect the role of DETCs (and indeed of other T lymphocyte classes; see below) by considering their morphological and functional heterogeneity [57,84,88,89], the variance of Skint1 expression across the population of mice studied (e.g., [55]), the relative speed of the DSCH elicitation phase in individual mice subjected or not to birth enucleation, and the characterization of the allergic phenotype of each mouse included in the study [90,91,92,93].

Since corneal nerve ablation induces the formation of contra-suppressor cells that disable T regulatory cells produced in the spleen after increasing the local availability of substance P (e.g., [9]), it might be that enucleation at birth alters systemic T regulatory cell recruitment after increasing substance P plasma levels. Our results show that bathing the auricular lymphoid node with an NK1 antagonist prevents the DSCH response in BE mice (Figure 4) supports this possibility. Further support comes from studies documenting that substance P (1) indeed circulates as a hormone in the bloodstream, (2) modulates systemic immune responses, (3) its plasma levels are altered in a variety of immunological conditions, and (4) exacerbates inflammation in different peripheral sites, including the skin [44,94,95]. Upcoming experiments must address the likelihood of this possibility.

### 3.3. Skin Langerhans’ Cells Migration to Lymph Nodes in S- and BE Mice

Langerhans cells, especially those in an immature state, are more likely to instruct inflammatory responses upon migration to lymph nodes [62,63,64,65,66,96,97,98]. We then evaluated whether fluorescein-sensitized Langerhans’ cells traveled to lymph nodes in greater numbers and/or at accelerated rates in 60-day-old BE mice when compared to their S counterparts, since reduced migration of these cells downsizes the contact hypersensitivity response [99]. CD11c^+^MHCII^+^CD207^+^/CD207^+^FITC^+^ Langerhans’ cells were sorted by flow cytometry, and their migration was recorded over four consecutive days (Figure 7a). Our results showed that the number of Langerhans cells moving from the ventral skin to the inguinal, brachial, and axillary lymph nodes varies greatly among individuals during the time window screened (Figure 7b). These results pinpoint that Langerhans’ cells migration is not perturbed by enucleation by birth under the experimental conditions tested. Nevertheless, because Langerhans’ cell trafficking may be differentially regulated depending upon the nature of the antigen used [100] and professional antigen-presenting cells, including Langerhans’, form a heterogeneous, cooperative, phenotypically diverse network in which each element has distinct life cycles, migratory properties and antigen-presenting functions [101,102,103], clearly our results fall short and are inconclusive. We think that the next experiments must test whether Langerhans cells migrate differentially to lymphoid organs following birth enucleation. 

## 4. Discussion

The eyes are best understood as visual organs. They, however, also instruct the immune system to become tolerant against foreign antigens on purpose, it is thought, to prevent eye autoimmune damage. The presence of frequent skin inflammatory diseases in young people with congenital or acquired (after birth) blindness [21,22] suggests, nonetheless, that the immune tolerance instructed by the eyes benefits other organs across the body. We then tested this premise by using delayed skin contact hypersensitivity (DSCH) as an experimental tool to induce and compare the inflammatory response developed by sighted (S) and birth-enucleated (BE) mice when sensitized and challenged with oxazolone or dinitrofluorobenzene at the ages of 10, 30 and 60 days of life. Fair enough, BE mice displayed an exacerbated, specific DSCH at 60 but not at 10 or 30 days of age. These results support the idea that the eyes may indeed prime systemic immune tolerance across the body. In agreement with this conclusion, the sensitization of the eye’s anterior chamber with a variety of antigens offers immunological protection against autoimmune diseases or reduces tissue damage after physical or vascular injury [104,105,106,107,108,109,110,111,112,113,114].

But when do the eyes instruct systemic immune tolerance? In mice, it is known that adult immune responses are conditioned early in postnatal development and that immune tolerance is set up by regulatory T lymphocytes also early in life [115]. So, it is possible that the systemic immune tolerance instructed by the eyes and the benefits it brings to the rest of the body are primed early in life. To empirically test this possibility, we surgically removed the eyes of neonatal mice before eye opening. Sixty-day-old BE mice consistently displayed an exacerbated DSCH. This response was unseen in BE mice at PD10 or PD30 and in mice enucleated at 60 days of age when tested at PD120. This set of observations supports the idea that the eyes condition systemic immune tolerance from early life and that this conditioning is long-lasting. Our presumptions gain support from observations showing that early postnatal enucleation increases liver macrophages’ phagocytic activity in adult rats [24] and that enucleation impairs the ability of the eye’s immune privilege to re-instate itself in mice subjected to experimental uveoretinitis [106].

The results commented on suggest that the eyes may condition the systemic immune response before eye opening. Although we did not define the timing when such conditioning happens, we think it might occur within the first ten days of life because, in rodents, this is the time window required for the blood–retinal barrier to be fully impermeable to foreign antigens [116,117,118,119]. Thus, between PD0 and PD10, the developing eyes might be exposed to a variety of antigens coming from different bodily sources. This early exposure could sensitize the ocular immune cells before migrating to secondary lymphoid organs, where they could instruct T regulatory memory lymphocytes, e.g., [106,120]. We presume that birth enucleation prevents this process. Though this notion is still a working model (Figure 8), it is known that (1) the immune system is instructed as early as a few hours after birth, even though immune maturity is reached only after the first month of life in rodents [27], (2) the adult immune response might be conditioned by neonatal immunological challenges as early as 3 to 5 days old (commented by [28]), and (3) the neonatal skin is colonized by regulatory T cells that mediate immune tolerance to commensal microbes [29]. All these circumstances might also explain why DSCH exacerbation is observed in BE mice until PD60. In any event, future experiments must address the premises and predictions of the proposed model.

At this point, we would like to comment on a few other physiological processes that might contribute to explaining DSCH exacerbation in 60-day-old BE mice. The recruitment of regulatory T lymphocytes is sensitive to circadian entrainment [128] and to melatonin’s systemic and local availability [129,130,131]. Neonatal enucleation might thus condition early colonization of the skin by regulatory T cells and/or resident memory T cells [27,132] after shifting the availability of melatonin [133]. In addition, birth enucleation might reduce local exposure to antigens derived from symbiotic/commensal microorganisms, thus disturbing the homeostasis of skin/mucosae immune tolerance [134,135,136,137,138,139]. Lastly, neonatal enucleation could prevent the eyes from providing direct instruction to immature T lymphocytes while passing through them [121,122].

A couple of considerations are worth making. In humans, previous epidemiological studies reported a higher-than-normal prevalence of transmissible and non-transmissible skin disorders among blind students [21,22]. Even though this can be attributed to poor personal hygiene habits, frequent skin scratching, overexposure to harsh climates and low living standards [21,22], our observations suggest that early blindness might compromise their systemic immune tolerance. In support of this possibility, human clinical studies show that oculocutaneous manifestations frequently precede multisystem inflammatory disorders [25,26]. In addition, mice (C57 black) that display underdeveloped anterior chambers or that lack the eye’s anterior segment have an increased frequency of bacterial infections [140]. In the absence of reliable data on the prevalence of associations between mouse skin inflammatory diseases and early or late-onset blindness, our results suggest that at least in the case of micro-ophthalmic C57 black mice, their increased susceptibility to develop inflammatory bacterial diseases might be due, in part, to the altered functional morphology of their eyes. 

## 5. Conclusions

The findings reported here suggest that the eyes provide information that conditions systemic immune tolerance before eye opening. The immune protection conferred by the eyes not only benefits themselves but the skin. These results also provide an alternative explanation for why early-blind youngsters and mice with eyes with altered development of the eye’s anterior chamber might have a greater prevalence of inflammatory skin disorders. Thus, upcoming experimental animal and human clinical studies must assess whether the systemic immunological tolerance of young blind subjects with a higher-than-average prevalence of inflammatory skin disorders is indeed impaired and whether adult individuals with late-onset blindness have immune tolerance relatively preserved. Also, future studies must use cell and tissue markers more closely associated with the development of immune tolerance in various body organs to begin establishing more directly the mechanistic framework underlying the eye-instructed systemic immune tolerance.

## Figures and Tables

**Figure 1 brainsci-13-01261-f001:**
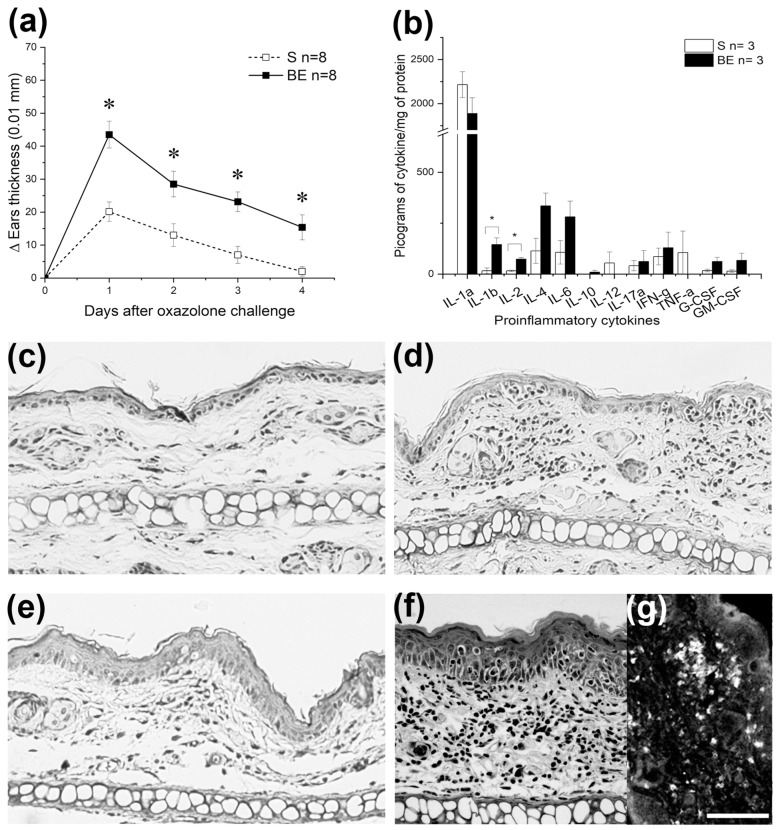
Adult mice enucleated at birth display an exacerbated delayed skin contact hypersensitivity (DSCH) response. (**a**) Time-lapse graph that depicts the progression of the ear’s inflammatory response observed in oxazolone (OXA-sensitized, sighted (S) or birth-enucleated (BE) 60-day-old mice after being challenged with OXA. Notice that DSCH is exacerbated after enucleation at birth. Data represents the mean ± SEM; Pair-wise Student’s *t*-test: * *p* < 0.05. (**b**) Bar graph that shows the concentration of pro-inflammatory cytokines in supernatants obtained from the ears of OXA-sensitized and challenged 60-day-old S or BE mice as determined by ELISA. The availability of proinflammatory cytokines tended to be higher in BE mice. Data represent the mean ± SEM; ANOVA test: * *p* < 0.05. Representative photomicrographs of ear sections of OXA-sensitized S mice stained with hematoxylin and eosin after being challenged with ethanol (**c**) or OXA (**d**). Representative photomicrographs of ear sections of OXA-sensitized BE mice stained with hematoxylin and eosin after being challenged with ethanol (**e**) or OXA (**f**). Notice that the lymphocytic infiltration occurs in OXA-challenged ears but not in ethanol-defied ones. Also, lymphocyte infiltrates are greater in BE mice. (**g**) A representative photomicrograph showing the presence of CD8^+^ lymphocytes in the infiltrates observed in OXA-sensitized and challenged BE mice, thus confirming that ear swelling is not a non-inflammatory edema. Scale bar = 30 μm. Data for (**a**,**b**) can be found in Appendix A.

**Figure 2 brainsci-13-01261-f002:**
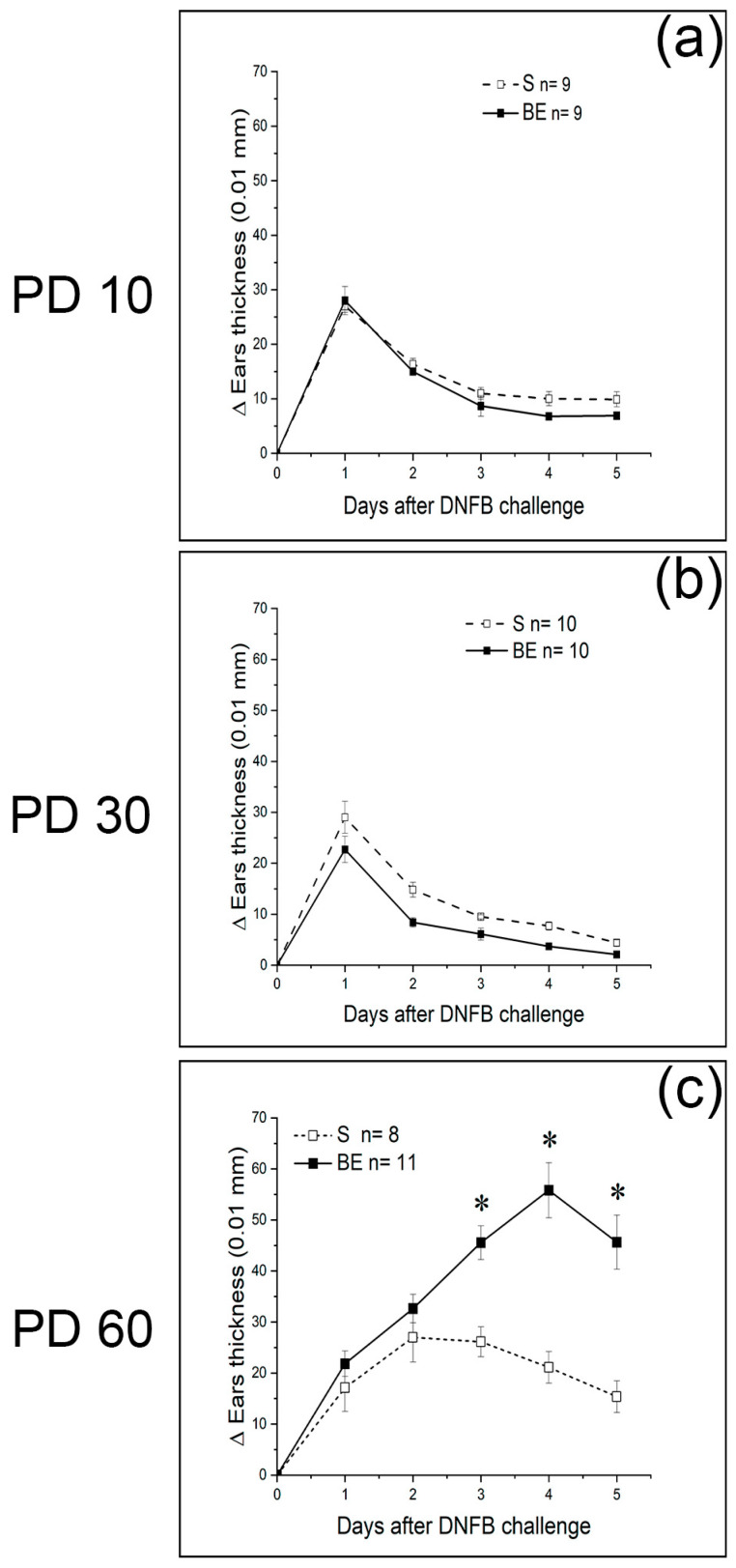
The eyes instruct systemic immune tolerance early after birth. Time-lapse graph that depicts the progression of the ear’s inflammatory response observed in dinitrofluorobenzene (DNFB) sensitized and challenged 10- (**a**) PD10), 30- ((**b**) PD30) and 60- ((**c**) PD60), day-old sighted (S) or birth enucleated (BE) mice. Note that, in BE mice, exacerbation of the delayed skin-contact hypersensitivity response occurs up to PD60. Data represent the mean ± SEM; Pair-wise Student’s *t*-test; * *p* < 0.05. Data for (**a**–**c**) can be found in Appendix A.

**Figure 3 brainsci-13-01261-f003:**
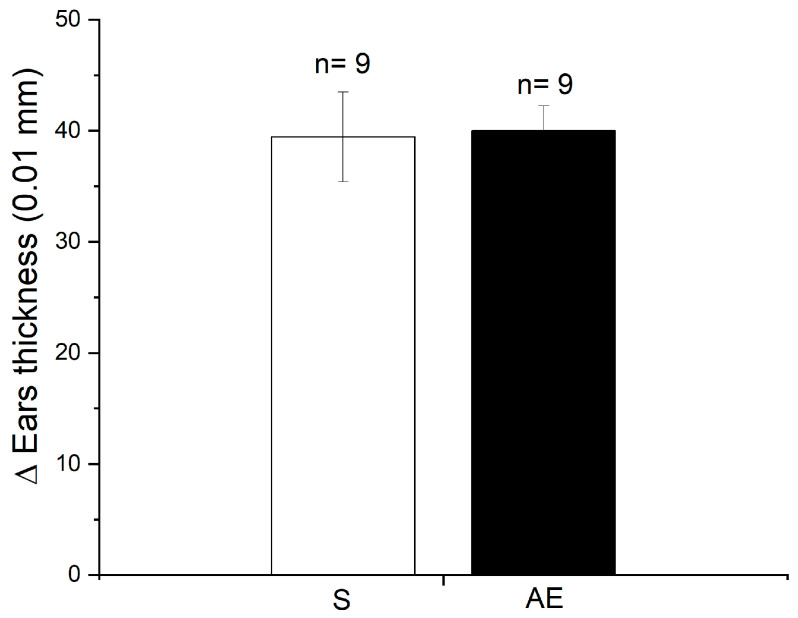
Mice enucleated at 60 days of age do not exhibit the exacerbation of the delayed skin contact hypersensitivity (DSCH) response when tested at 120 days of age. Bar graph that depicts the magnitude of the skin contact hypersensitivity response in oxazolone (OXA) sensitized and challenged, 120-day-old sighted (S) or enucleated mice. In the latter group of mice, enucleation was performed at 60 days of age. Notice that both S and adult enucleated (AE) mice’s ears inflame to a similar extent, relative to the thickness of the contralateral ears that were brushstroked with ethanol only. Thus, adult enucleation does not cause DSCH to overreact as it does after neonatal enucleation. Data represent the mean ± SEM; Pair-wise Student’s *t*-test: non-significant. Data for this figure can be found in Appendix A.

**Figure 4 brainsci-13-01261-f004:**
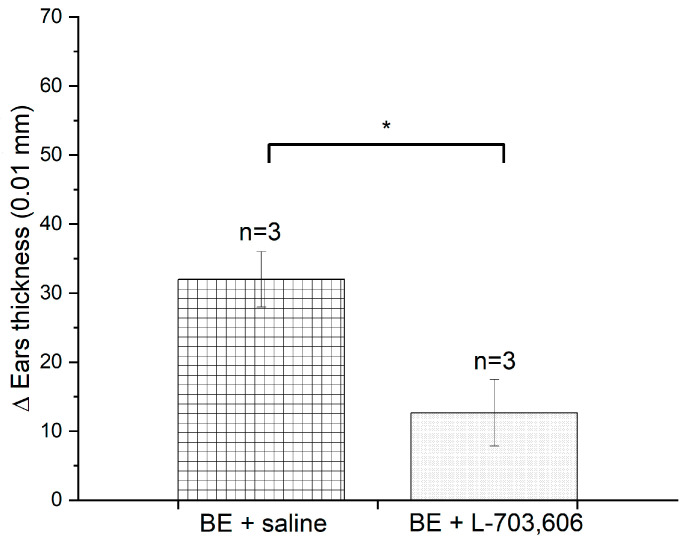
The antagonization of substance P prevents the exacerbation of delayed skin contact hypersensitivity (DSCH) responses in birth-enucleated (BE) mice. Bar graph that depicts the magnitude of DSCH in oxazolone (OXA) sensitized and challenged 60-day-old BE mice following the irrigation of the auricular lymph node with L-703,606, a drug that antagonizes substance P binding to the NK-1 receptor. Note that L-703,606 prevents DSCH from progressing after the OXA challenge, suggesting that effector T lymphocyte migration and recruitment are hindered and that efferent neural mechanisms are involved. Data represent the mean ± SEM; Pair-wise Student’s *t*-test; * *p* < 0.05. Data for this figure can be found in Appendix A.

**Figure 5 brainsci-13-01261-f005:**
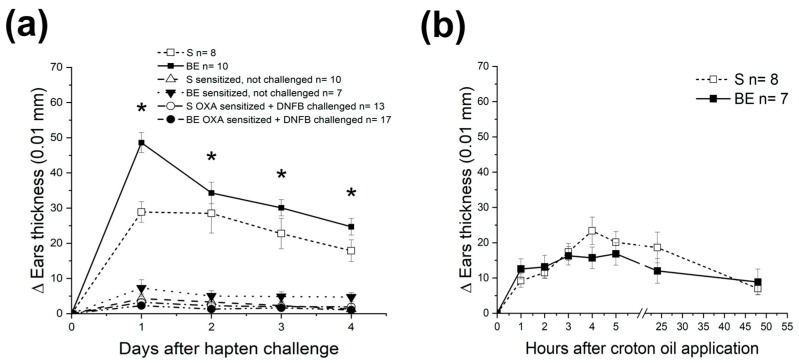
Delayed skin contact hypersensitivity (DSCH) responses, whether exacerbated or not, are hapten-specific. (**a**) Time-lapse graph that depicts the progression of the ear’s inflammatory response observed in oxazolone (OXA) sensitized and challenged 60-day-old sighted (S) or birth enucleated (BE) mice. The graph also shows the progression of DSCH in 60-day-old S and BE mice that were OXA sensitized but unchallenged or OXA sensitized but challenged with dinitrofluorobenzene (DNFB). Notice that none of the last four groups developed DSCH. (**b**) Time-lapse graph that depicts the progression of the ear’s inflammatory response observed in croton oil-sensitized and challenged 60-day-old S or BE mice. Croton oil alone elicits a long-lasting irritant response of similar magnitude in both groups of mice. Data represent the mean ± SEM; Pair-wise Student’s *t*-test: * *p* < 0.05. Data for this figure can be found in Appendix A.

**Figure 6 brainsci-13-01261-f006:**
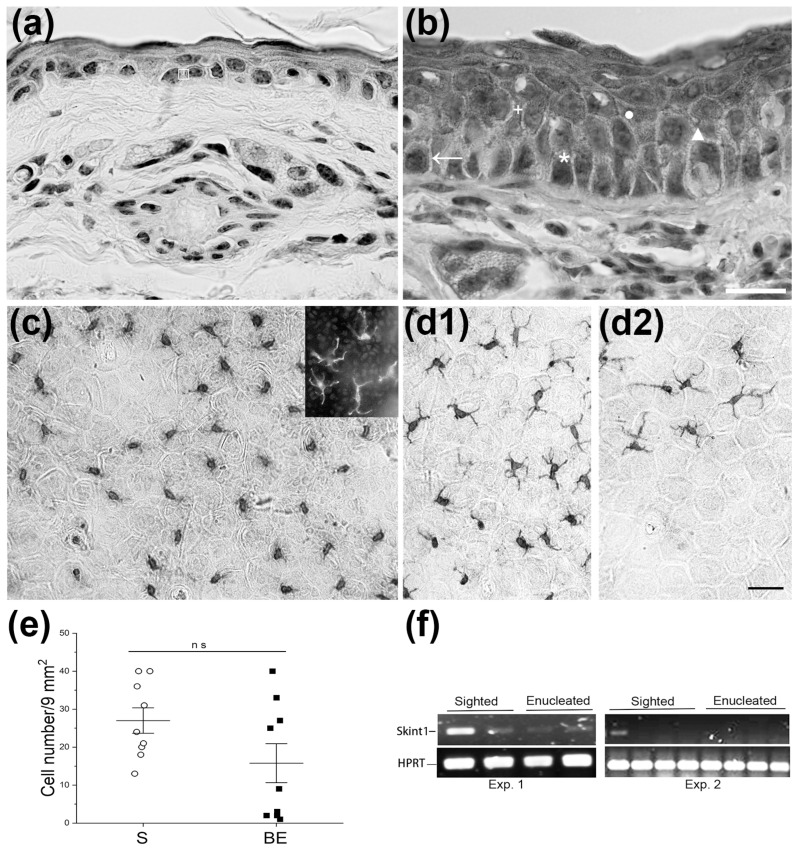
Neonatal enucleation modifies the cytoarchitecture of the epidermis. Representative photomicrographs of ear skin sections obtained from sighted (**a**) S) and birth-enucleated (**b**) BE) mice stained with hematoxylin and eosin. Keratinocytes become hypertrophied and the epidermis becomes multilayered in 60-day-old BE mice. Square indicates basal cell layer cuboidal cell in (**a**), arrowhead denotes structures resembling spines between adjacent cells of the stratum *germinativum*, cross and bullet designate stratum *spinosum* cells, asterisk shows stratum *germinativum* columnar cell and left vertex of the triangle points to intercellular connections between stratum *spinosum* cells in (**b**). Scale bar = 20 μm for (**a**,**b**). Representative photomicrographs of epidermal sheets of 60-day-old S ((**c**) and BE ((**d1**,**d2**)) mice immunostained against CD3ε. At first glance, S mice seem to display similar numbers of DETCs across individuals (**c**). In contrast, BE mice displayed two phenotypes, with some having greater DETC numbers (**d1**) than others (**d2**). The inset in (**c**) shows that CD3ε positive cells displayed Vγ5-TCR immunoreactivity, thus suggesting that these cells are descendants of those colonizing the skin from the thymus likely during prenatal life. Scale bar = 30 μm for (**c**,**d1**,**d2**). (**e**) Column scatter graph shows the density of DETCs in epidermal sheets across 60-day-old S or BE mouse populations. Two mouse phenotypes are observed in both groups: those having DETC densities above or below average. The difference between phenotypes sharpens after neonatal enucleation. Data represent the mean ± SEM; Pair-wise Student’s *t*-test: ns, non-significant. (**f**) Digital photographs that show the results obtained after amplifying Skint1 mRNA isolated from the skin of the ears of six, 60-day-old S or BE mice by using RT-PCR. Skint1 mRNA was readily amplified in two out of six S mice (first lane in (**f**) exp1 and (**f**) exp2) and weakly amplified in one out of six S mice (second lane in (**f**) exp1) and in two out of six BE mice (third and fourth lanes in (**f**) exp1). Data for (**e**) can be found in Appendix A.

**Figure 7 brainsci-13-01261-f007:**
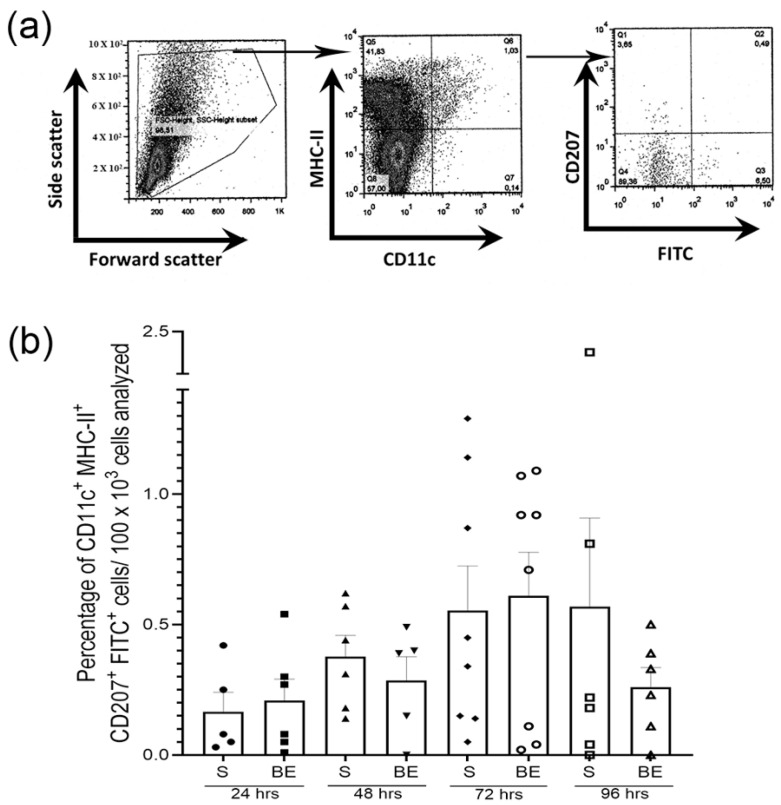
Neither the relative quantity (i.e., event number at a given time point) nor the temporal pattern of migration of epidermal Langerhans’ cells recovered from draining lymph nodes differed between 60-day-old sighted (S) and birth enucleated (BE) mice. (**a**) Representative flow cytometry dot plots that delineate the scatter parameters and gating strategy used in the workflow sequence employed to sort and isolate CD11c^+^ MHCII^+^ CD207^+^ FITC^+^ Langerhans’ cells in both adult fluorescein-sensitized S and BE mice. This example comes from an experiment conducted on an S mouse. (**b**) Bar graph that documents the percentage of CD11c^+^ MHCII^+^ CD207^+^ FITC^+^ Langerhans’ cells per 100K cells obtained from pulls of the inguinal, brachial, and axillary lymph nodes in fluorescein-sensitized 60-day-old S or BE mice over time. Each geometric figure (circles, squares, triangles and romboids) spreading along the axis of each column represents individual mice per group at different time points. Data represent the mean ± SEM; Pair-wise Student’s *t*-test: non-significant. Data for (**b**) can be found in Appendix A.

**Figure 8 brainsci-13-01261-f008:**
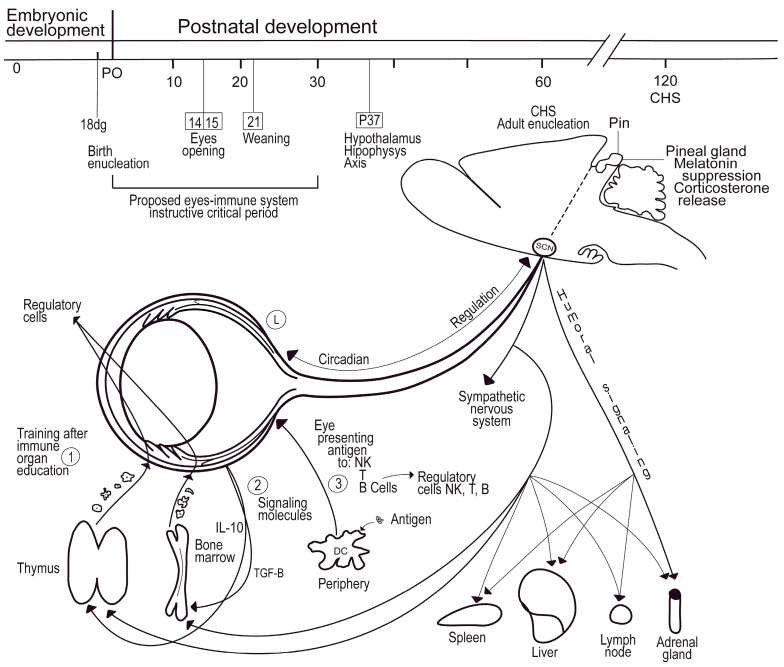
Illustration of the putative mechanism that underlies the formation of body immune tolerance under the instruction of the eyes during early postnatal life. The eyes would instruct the body’s immune tolerance from birth to several days after eye opening (1). We presumed that, during the first ten to thirty days after birth, bone marrow or thymus-educated leucocytes might reach the eyes, where they could receive complementary instructions on immune tolerance. Indeed, the eyes may operate as a surrogate home for developing T lymphocytes to become fully functional [121]. (2) Neonatal eyes could provide soluble messengers (e.g., TGF-β, αMHS, IL-10; [40,122]) or already-instructed cells [13,18,20] that, after reaching the bone marrow and/or thymus, may help refine the development of immune cells in these locales. (3) Peripheral antigen-presenting cells may visit the eyes prior to blood–retinal barrier formation. While doing so, they might display antigens that help the eyes instruct their own regulatory immune cells. (L) The inflammatory response could be primed by light-independent or light-dependent circadian rhythms [123,124], even if the eyes are not open and the eyelids are translucent. Neuroendocrine signals may also participate in the regulation of these processes [125,126,127].

## Data Availability

Databases are made available as Appendix A.

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
