# Peer review of "Looking beyond Self-Protection: The Eyes Instruct Systemic Immune Tolerance Early in Life"

_brainsci, 2023, doi:10.3390/brainsci13091261_

Round 1

Reviewer 1 Report

Looking beyond self-protection: eyes instruct systemic immune tolerance early in life.

The study provides valuable clinical information about the eyes instructing systemic immune tolerance early in life.

Eye-instructed immune tolerance is supposed to provide self-protection. The frequent inflammatory skin diseases that young blind people develop suggest that the eyes train an immune tolerance that benefits the whole body. The authors use the chemical compounds oxazolone or dinitrofluorobenzene of the ketones functional group to assess immunological activity with the DTH (delayed hypersensitivity) test.

The eyes are known to instruct systemic immune tolerance that benefits organs outside the eyes from an early age. The higher prevalence of reported inflammatory skin disorders in the young might reflect reduced immunological tolerance associated with altered functional morphology of the eyes.

However, the work has some limitations, such as in the bibliographic reviews, that is, the references of 123 cited, there are (5) citations in 2019, (6) in 2020, (5) in 2021, only 1 in 2022 and no citations cited in the present year{o 2023, I believe that the authors should update their review of the literature if they want their manuscript to be accepted. Reading and reviewing the manuscript, I note that the authors did not discuss very well the prevalence of inflammatory skin disorders reported in animal studies (the difference between young and adult animals) as it may reflect reduced immunological tolerance associated with the altered functional morphology of the eyes. They did not present studies on inflammatory processes evaluating cytokines and comparing humoral and cellular responses (B lymphocytes and T lymphocytes) as well as TLR tele receptors.

Reviewing the literature carried out by the authors, I observe that there is a great lack in which I recommend to the authors of the manuscript to increase the updated references.

 In relation to these observations, I have a few questions:

1) What is the relationship between the eyes and systemic immunological tolerance at a stage of life?

2) Why few proteins can be used as potential biomarkers in systemic immune tolerance?

3) Can you explain the relationship between the pathophysiology of the eye and inflammation markers at different stages of life and how it affects systemic immunological tolerance?

4) Some clinical markers are quite non-specific and refer mainly to the anatomical/morphological characteristics in the eye in terms of immunological tolerance. In this sense, what do you suggest?

5) How can one explain the skin inflammation in 60-day old BE mice was hapten specific and supported by incoming CD8+ lymphocytes?

Minor editing of English language required

Author Response

Enclosed, please find the revised version of the manuscript entitled “Looking beyond self-protection: The eyes instruct systemic immune tolerance early in life”. We have addressed all the suggestions, comments and criticisms made by both reviewers. Accordingly, through this letter, we comment the changes introduced to the text based on both reviewers’ opinions. We thank the reviewers for their time and contributions since the article’s text and layout improved significantly after considering their comments. We also apologize with the editorial team and reviewers for the time taken to get back to you all. We intended to cover fully the review of the literature requested by reviewer #1. We hope we interpreted such a request correctly. After these introductory remarks, we now pass to the body of the response letter. In so doing, we strategically decided to break up our responses into sections. We believe that, in this manner, we could be more explicit, and express clearer, the modifications made to the article’s narrative after considering reviewers’ #1 and #2 objections, suggestions, additions and comments.

Reviewer (R) #1

 Comment:

“The study provides valuable clinical information about the eyes instructing systemic immune tolerance early in life. Eye-instructed immune tolerance is supposed to provide self-protection. The frequent inflammatory skin diseases that young blind people develop suggest that the eyes train an immune tolerance that benefits the whole body. The authors use the chemical compounds oxazolone or dinitrofluorobenzene of the ketones functional group to assess immunological activity with the DTH (delayed hypersensitivity) test”. The eyes are known to instruct systemic immune tolerance that benefits organs outside the eyes from an early age. The higher prevalence of reported inflammatory skin disorders in the young might reflect reduced immunological tolerance associated with altered functional morphology of the eyes”.

Response:

We agree with R#1 about the overall layout of our study. We would like to remark, however, that this work was designed primarily to test two previously unexplored premises the first of which states that the eyes help preventing the immune system to overreact through the body (this in fact was not known, it was at best expected), and not exclusively within the eyes themselves as it has been previously established (see references 3-6, 9-14, 17-20). In other words, it is an accepted fact -not supposed- that the eyes provide immune tolerance to protect themselves from immunological damage. Here, we now provide evidence that supports that the eyes indeed help instructing immunotolerance across the body (i.e., one of the premises to be evaluated), that is beyond eye boundaries, even though the mechanism remains unclear. In addition, our study provides sufficient circumstantial evidence to support our second premise that sustains that the ability of the eyes to induce body-wide immune tolerance is time sensitive; the instruction likely takes place within the first few days-weeks after birth, lasting well into adulthood even if the eyes are lost at adult age.

We would also like to clarify that our study may indeed have clinical implications. At this stage, however, our study provides basic scientific information, obtained in mice, on the role of the eyes in the regulation of systemic immune tolerance, and on the time window when the eyes immune instruction might take place. Clearly, both premises must be corroborated by clinical studies conducted in children and adults (likely in other animal species) with early and late onset blindness associated with diverse causes, before advancing further thoughts on human clinical implications or usage. In addition, according to NIH, a clinical research study is defined as the one “in which one or more human subjects are prospectively assigned to one or more interventions (which may include placebo or other control) to evaluate the effects of those interventions on health-related biomedical or behavioral outcomes” https://grants.nih.gov/policy/clinical-trials/definition.htm). For this reason, we believe our study must be considered as a contribution of basic scientific research with clinical implications, but not yet as a work providing clinical information that could, in the short run, be used to understand naturally occurring mouse and/or human skin diseases. For the moment, at best, our study “provides room” for a limited set of speculations / implications (commented in lines 580-591).

Lastly, we want to emphasize that DSCH was used as an experimental animal unit to model a physiological condition to help us testing the premises already commented. Our study was not intended to replicate clinical conditions whose physiopathology was to be studied or solved. The clinical circumstances referred in the text about young blind people allowed us to infer the premises to be explored, and to set the logic of the arguments used to conduct the experiments reported.    

Hence, as a natural consequence of the issues wisely commented by R#1, we try to make both matters clear by adding assertive words/phrases (underlined and highlighted in yellow) whenever we consider it convenient throughout the abstract (lines 16-20), introduction (lines 33 and 34; 38-42), discussion (lines 506-519; 525; 582-593) and conclusion (lines 597-603) sections.

Comment:

“However, the work has some limitations, such as in the bibliographic reviews, that is, the references of 123 cited, there are (5) citations in 2019, (6) in 2020, (5) in 2021, only 1 in 2022 and no citations cited in the present year 2023, I believe that the authors should update their review of the literature if they want their manuscript to be accepted. Reading and reviewing the manuscript, I note that the authors did not discuss very well the prevalence of inflammatory skin disorders reported in animal studies (the difference between young and adult animals) as it may reflect reduced immunological tolerance with the altered functional morphology of the eyes. They did not present studies on inflammatory processes evaluating cytokines and comparing humoral and cellular responses (B lymphocytes and T lymphocytes) as well as TLR tele receptors. Reviewing the literature carried out by the authors, I observe that there is a great lack in which I recommend to the authors of the manuscript to increase the updated references.”

Response

In following R#1 advice, we conducted a systematic search of the literature (2019-2023) by using the following sentences “prevalence of inflammatory skin disorders reported in animal studies (the difference between young and adult animals) as it may reflect reduced immunological tolerance associated with the altered functional morphology of the eyes” and “present studies on inflammatory processes evaluating cytokines and comparing humoral and cellular responses (B lymphocytes and T lymphocytes) as well as TLR tele receptors” as guides to create numerous search entries based on a network of matrices fabricated by combining all of the searching terms considered. The following are just a handful of examples of the arrangements used to conduct the searches:

Prevalence AND/OR inflammatory skin disorders AND/OR mice AND/OR immune tolerance AND/OR enucleation

Prevalence AND/OR inflammatory skin disorders AND/OR mouse AND/OR immune tolerance AND/OR enucleation

Prevalence AND/OR inflammatory skin disorders AND/OR mice AND/OR immune tolerance AND/OR blindness

Prevalence AND/OR inflammatory skin disorders AND/OR mouse AND/OR immune tolerance AND/OR enucleation

Prevalence AND/OR inflammatory skin disorders AND/OR mice AND/OR immune tolerance AND/OR visual deprivation

Prevalence AND/OR inflammatory skin disorders AND/OR mouse AND/OR immune tolerance AND/OR visual deprivation

Prevalence AND/OR inflammatory skin disorders AND/OR mice AND/OR immune tolerance AND/OR altered vision

Prevalence AND/OR inflammatory skin disorders AND/OR mouse AND/OR immune tolerance AND/OR altered vision

Prevalence AND/OR inflammatory skin disorders AND/OR mice AND/OR immune tolerance AND/OR impaired vision

Prevalence AND/OR inflammatory skin disorders AND/OR mouse AND/OR immune tolerance AND/OR impaired vision

Prevalence AND/OR inflammatory skin disorders AND/OR mice AND/OR immune tolerance AND/OR microphthalmia

Prevalence AND/OR inflammatory skin disorders AND/OR mouse AND/OR immune tolerance AND/OR microphthalmia

Prevalence AND/OR inflammatory skin disorders AND/OR mice AND/OR immune tolerance AND/OR blindness

Prevalence AND/OR inflammatory skin disorders AND/OR mouse AND/OR immune tolerance AND/OR blindness

In all these entries we also incorporated and combined in various forms the terms:

  • AND/OR adult mouse or mice and AND/OR young mouse or mice instead of mouse or mice alone.
  • AND/OR cytokines, AND/OR B lymphocytes, AND/OR T lymphocytes, AND/OR Toll-Like AND/OR Toll-like Receptor Signaling Pathways
  • AND/OR early onset blindness and AND/OR late onset blindness

We carefully revised the bibliography listed under each of the search entries used when tested in the following search engines/data bases: BIOLOGICAL ABSTRACTS (1993-actual), EBSCOHOST, PROQUEST, PUBMED, SCOPUS, WEB OF SCIENCE, ACADEMIA.EDU, MICROSOFT ACADEMIC, SCHOLARPEDIA.ORG WORLDWIDESCIENCE.ORG, CURRENT CONTEST CONNECT, CAB ABSTRACTS, GOOGLE SCHOLAR, and RESEARCH GATE. Finally, we also assessed CLINICALTRIALS.GOV to identify human studies with a translational context and for patents using DERWENT INNOVATION INDEX.

Unfortunately, after almost a month of pursuing this task, we could not identify original / review publications, patents or clinical trials matching the information requirements requested by R#1, thus suggesting that our work might be quite original with virtually no direct extant antecedents published. About the prevalence of skin inflammatory disorders in mice, we could not find data on the matter. The literature makes clear, however, that the mouse is the species most frequently used to approach DSCH to understand human atopic dermatitis, psoriasis, and other skin inflammatory diseases (Material and Methods lines 129-131). Despite the unsuccessful attempt, we introduced to the text and added to the reference list, some references (2019-2023) we thought have inferential implications to our work and findings. Such references were incorporated to and highlighted throughout the text and at the very end of the reference list to make them easier to find. Given the results, no major additional paragraphs / sections were introduced to the original manuscript (Discussion: lines 580-590; Conclusions: 590-601). We hope R#1 does not feel ignored. Quite on the contrary, we did our best effort to inquire and address what R#1 quite correctly detected. We hope our searching effort is well appraised by R#1. Of course, if R#1 knowns specific literature of interest that we are still missing, we are completely willing to introduce and discuss the articles R#1 might find useful.

The articles introduced to and highlighted through the text are listed below:   

Scheinman, P.L.; Vocanson, M.; Thyssen, J.P.; Johansen, J.D.; Nixon, R.L.; Dear, K.; Botto, N.C.; Morot, J.; Goldminz, A.M. Contact dermatitis. Nat Rev Dis Primers. 2021, 7, 38. doi:10.1038/s41572-021-00271-4

McGraw, J.M; Witherden, D.A. γδ T cell costimulatory ligands in antitumor immunity. Explor Immunol. 2022, 2, 79–97. doi:10.37349/ei.2022.00038

Herrmann, T; Karunakaran, M.M. Butyrophilins: γδ T Cell Receptor Ligands, Immunomodulators and More. Front Immunol. 2022, 13, 876493. doi: 10.3389/fimmu.2022.876493.

Tramontana, N.; Hansel, K.; Bianchi, L.; Sensini, C.; Malatesta, N.; Stingeni, L. Advancing the understanding of allergic contact dermatitis: from pathophysiology to novel therapeutic approaches. Front. Med. 2023, 10, 1184289. doi: 10.3389/fmed.2023.1184289

Brand, A; Hovav, A-H; Clausen, B.E. Langerhans cells in the skin and oral mucosa: Brothers in arms? Eur J Immunol. 2023, 53, e2149499. doi: 10.1002/eji.202149499

Talagas, M. Anatomical contacts between sensory neurons and epidermal cells: an unrecognized anatomical network for neuro-immuno-cutaneous crosstalk. Br J Dermatol. 2023, 188, 176–185. doi:10.1093/bjd/ljac066

Peterson, P.; Kisand, K.; Kluger, N.; Ranki, A. Loss of AIRE-mediated immune tolerance and the skin. J Invest Dermatol. 2022, 142, 760-767. doi: 10.1016/j.jid.2021.04.022

Chen, R.; Routh, BN.; Gaudet, A.D.; Fonken, L.K. Circadian Regulation of the Neuroimmune Environment Across the Lifespan: From Brain Development to Aging. J Biol Rhythms. 2023, 0. doi:10.1177/07487304231178950

Jerigova, V.; Zeman, M.; Okuliarova, M. Circadian Disruption and Consequences on Innate Immunity and Inflammatory Response. Int. J. Mol. Sci. 2022, 23, 13722. doi: 10.3390/ijms232213722

Mueller, S.N. Neural control of immune cell trafficking. J Exp Med. 2022, 219, e20211604. doi: 10.1084/jem.20211604

Liu, A.W.; Gillis, J.E.; Sumpter, T.L.; Kaplan, D.H. Neuroimmune interactions in atopic and allergic contact dermatitis. J Allergy Clin Immunol. 2023, 151, 1169-1177. doi: 10.1016/j.jaci.2023.03.013.

Marek-Jozefowicz, L.; Nedoszytko, B.; Grochocka, M.; Żmijewski, M.A.; Czajkowski, R.; Cubała, W.J.; Slominski, A.T. Molecular Mechanisms of Neurogenic Inflammation of the Skin. Int. J. Mol. Sci. 2023, 24, 5001. doi: 10.3390/ijms24055001

Finally, regarding “They did not present studies on inflammatory processes evaluating cytokines and comparing humoral and cellular responses (B lymphocytes and T lymphocytes) as well as TLR tele receptors”, we think this objection is to some extent overcome, based upon our findings, in the complementary discussion included along the Results (lines 414-450) and Discussion (Figure 8 and its figure legend; added to the revised version) sections.

Comment:

“In relation to these observations, I have a few questions:

“1) What is the relationship between the eyes and systemic immunological tolerance at a stage of life?”

Response:

Based upon our results, we think that the presence of eyes with normal functional morphology by birth and up to the first month of life, assures the ability of the eyes to instruct systemic immune tolerance. If the normal eyes are present until the animal reaches adult age, eye induced immune tolerance becomes permanent, and the eyes are no longer needed to maintain it (Introduction lines 59-61).

Comment:

“2) Why few proteins can be used as potential biomarkers in systemic immune tolerance?”

Response:

In this work we used the markers (CD4+, CD8+, MHC-II, Cd11c, Langerin, SkinT1, CD3+, NK1) which we believe (budget limitations) would help to study the dynamics of immune cells during DSCH under the assessed experimental conditions. We recognize that such markers are not specific to follow de development of systemic immune tolerance. We are just at the stage of exploring the premises commented previously. It is therefore important for us to explicitly recognize the limitations of our work base upon this comment; please, refer to the Conclusion section (lines 604-606). In this regard, there are other biomarkers such as FoxP3, Aire, RORgt, TGFb, CD80, CD86, PDL1,2, Zbtb, CCR7, Vista, CD28, CTLA-4, Ox40 and Ox40L that can be used to pursue this quest.

Comment:

3) Can you explain the relationship between the pathophysiology of the eye and inflammation markers at different stages of life and how it affects systemic immunological tolerance?”

Response:

The short answer is we cannot. Since 1) this was an exploratory, descriptive work, 2) we have used a small number of phenotypic markers to identify a handful of immune cells potentially involved in DSCH, and 3) such markers were only used in a single time point (60 days of age), it is impossible for us to explain the functional relationship alluded. This is a limitation of our work that has been explicitly recognized in the text (Conclusion lines 604-607). Despite this fact, we intended to approach this very important question by proposing a conceptual model schematized in Figure 8. Please, also refer to Discussion sections (lines 570-579)

Comment:

4) Some clinical markers are quite non-specific and refer mainly to the anatomical/morphological characteristics in the eye in terms of immunological tolerance. In this sense, what do you suggest?

Response:

As mentioned before, this is an unavoidable limitation of the present work, a pitfall that has been explicitly recognized in the conclusion section (lines 604-606). This drawback may be overcome by using, in more focused future studies, numerous other markers specifically associated with the development of systemic immune tolerance.

Comment:

5) How can one explain the skin inflammation in 60-day old BE mice was hapten specific and supported by incoming CD8+ lymphocytes?”

Response:

The phrase is certainly confusing, and it has been rephrased. It now reads “Skin inflammation in 60-day-old, BE mice was hapten exclusive and supported by distinct CD8+ lymphocytes (Abstract lines 23-24). It has been previously shown that CD8+ T lymphocytes are the cells that oversee the proinflammatory response of the elicitation phase during contact hypersensitivity. They are one of the first immune cell types to arrive when the skin is challenged under DSCH conditions (Gorbachev, A.V.; Fairchild, R.L. CD4+ T Cells Regulate CD8+ T cell-mediated cutaneous immune responses by restricting effector T cell development through a Fas ligand-dependent mechanism. J Immunol 2004, 172, 2286-2295. doi: 10.4049/jimmunol.172.4.2286). Because, during the elicitation phase, there was no cross inflammatory responses to DNFB and oxazolone, and early lymphocytic infiltration was CD8+ in both cases, we inferred that DSCH response for either antigen was supported by “distinct CD8+ lymphocytes”. We hope the rephrasing helps clearing the confusion.

Reviewer 2 Report

I want to congratulate the authors for their amazing work. Your study utilizing mouse models has shed light on the intriguing aspect of immune regulation that has significant implications for the scientific community at large. The clarity with which you communicated your findings ensures that fellow researchers can readily comprehend and build upon your work, facilitating collaborative research and accelerating progress in this important area of research. 

I had some minor queries:

1. Line 510-512: "The presence of frequent inflammatory skin diseases in young blind people suggests, nonetheless, that the immune tolerance instructed by the eyes benefits other organs across the body." Please provide citations.

2. Line 555-557: "Previous epidemiological studies reported higher than normal prevalence of transmissible and non-transmissible skin disorders among blind students." 

What was the cause of blindness?

Does enucleation matter or any cause of bilateral blindness is associated with inflammatory skin disorders?

3. Line 530-532: "We think it might occur within the first ten days of life because, in rodents, this is the time window required for the blood-retinal barrier to be fully impermeable to foreign antigens." Is there a similar window for human eyes?

4. Is there some recommendation by the authors to conduct a study in human beings? Something like comparing inflammatory skin disorders in bilateral anophthalmic patients vs age-matched controls?

Author Response

Enclosed, please find the revised version of the manuscript entitled “Looking beyond self-protection: The eyes instruct systemic immune tolerance early in life”. We have addressed all the suggestions, comments and criticisms made by both reviewers. Accordingly, through this letter, we comment the changes introduced to the text based on both reviewers’ opinions. We thank the reviewers for their time and contributions since the article’s text and layout improved significantly after considering their comments. We also apologize with the editorial team and reviewers for the time taken to get back to you all. We intended to cover fully the review of the literature requested by reviewer #1. We hope we interpreted such a request correctly. After these introductory remarks, we now pass to the body of the response letter. In so doing, we strategically decided to break up our responses into sections. We believe that, in this manner, we could be more explicit, and express clearer, the modifications made to the article’s narrative after considering reviewers’ #1 and #2 objections, suggestions, additions and comments.

Reviewer (R) #2

Comment:

I want to congratulate the authors for their amazing work. Your study utilizing mouse models has shed light on the intriguing aspect of immune regulation that has significant implications for the scientific community at large. The clarity with which you communicated your findings ensures that fellow researchers can readily comprehend and build upon your work, facilitating collaborative research and accelerating progress in this important area of research. 

Response:

Authors thank the appreciation and encouraging words.

Comment:

I had some minor queries:

“1. Line 510-512:

“The presence of frequent inflammatory skin diseases in young blind people suggests, nonetheless, that the immune tolerance instructed by the eyes benefits other organs across the body." Please provide citations.”

Response:

References have been provided. Please see lines 508-511 in the Discussion section. Notice that the text after the comma… “suggests, nonetheless, that the immune tolerance instructed by the eyes benefits other organs across the body” … is one of the premises explored in our work. Therefore, no reference exists to support this statement but the present results.

Abolfotouh, M.A.; Bahamdan, K. Skin disorders among blind and deaf male students in Southwestern Saudi Arabia. Ann. Saudi Med. 2000, 20, 161-164. Reference 21 in the text.

Fathy, H.; El-Mongy, S.; Baker, N.I.; Abdel-Azim, Z.; El-Gilany, A. Prevalence of skin diseases among students with disabilities in Mansoura, Egypt. East. Mediterr. Health J. 2004, 10, 416-424. Reference 22 in the text.

“2. Line 555-557:

"Previous epidemiological studies reported higher than normal prevalence of transmissible and non-transmissible skin disorders among blind students." 

“What was the cause of blindness?"

Response:

Unfortunately, authors did not describe the etiology. They only refer blindness to be congenital or acquired after birth following fever episodes, infections, accidents, or surgeries. This information has been stated in the text. Please refer to lines 40-43 in the Introduction section and lines 508 and 509 in the Discussion section.

"Does enucleation matter or any cause of bilateral blindness is associated with inflammatory skin disorders?"

Response:

From our experience, brain plasticity (Fetter-Pruneda I, Geovannini-Acuña H, Santiago C, et al. Shifts in developmental timing, and not increased levels of experience-dependent neuronal activity, promote barrel expansion in the primary somatosensory cortex of rats enucleated at birth. Plos one. 2013 ;8(1):e54940. DOI: 10.1371/journal.pone.0054940. PMID: 23372796; PMCID: PMC3556040) and skin plasticity (see Figure 6 in our manuscript) requires an extensive functional anatomical damage of both eyes. So, we would say that any condition that really alters the functional morphology of the eyes bilaterally is required to decrease immune tolerance.   Clearly, unilateral enucleation is a condition worth to test.

Comment:

“3. Line 530-532:

"We think it might occur within the first ten days of life because, in rodents, this is the time window required for the blood-retinal barrier to be fully impermeable to foreign antigens." Is there a similar window for human eyes?”

Response:

To our knowledge, there are no published data documenting a similar window for the human eyes. A recent review, however, suggest that such a period may exist (3-6 months of life; Donald, K.; Finlay, B.B. Early-life interactions between the microbiota and immune system: impact on immune system development and atopic disease. Nat Rev Immunol, 2023. https://doi.org/10.1038/s41577-023-00874-w)

 Comment:

“4. Is there some recommendation by the authors to conduct a study in human beings? Something like comparing inflammatory skin disorders in bilateral anophthalmic patient’s vs age-matched controls?”

Response:

The recommendation, after the temptation induced by R#2, is disclosed in lines 61-65 and 600-603 in the Introduction and Conclusion sections.